# No Soundness in the Real World:
# On the Challenges of the Verification of Deployed Neural Networks

**Attila Szász** [1]   **Balázs Bánhelyi** [1]   **Márk Jelasity** [1,2]

## Abstract

The ultimate goal of verification is to guarantee the safety of *deployed* neural networks. Here, we claim that all the state-of-the-art verifiers we are aware of fail to reach this goal. Our key insight is that *theoretical soundness* (bounding the full-precision output while computing with floating point) does not imply *practical soundness* (bounding the floating point output in a potentially stochastic environment). We prove this observation for the approaches that are currently used to achieve provable theoretical soundness, such as interval analysis and its variants. We also argue that achieving practical soundness is significantly harder computationally. We support our claims empirically as well by evaluating several well-known verification methods. To mislead the verifiers, we create adversarial networks that detect and exploit features of the deployment environment, such as the order and precision of floating point operations. We demonstrate that all the tested verifiers are vulnerable to our new deployment-specific attacks, which proves that they are not practically sound.

## 1. Introduction

The formal verification of an artificial neural network produces a mathematical proof that the network has (or does not have) a certain important property. One important property to verify is the adversarial robustness (Szegedy et al., 2014) of classifier networks, where we wish to prove that a subset of inputs are all assigned the same class label.

There is a wide variety of approaches that address this problem (see Section 2). However, these methods invariably focus on the verification of the theoretical model of the network, that is, they seek to characterize the behavior of the full-precision computation. Most solutions carefully address floating point issues as well, but only as an obstacle in the way of verifying the theoretical model, e.g. (Singh et al., 2019a; 2018).

At the same time, deployment environments affect numeric computation through hardware and software features, often resulting in stochastic behavior (Schlögl et al., 2023; Villa et al., 2009; Shanmugavelu et al., 2024). This means that the verification of the theoretical model and the verification of the deployed network are different problems. Parallel to our work, a similar observation was also made by (Cordeiro et al., 2025), where this problem is referred to as the *implementation gap*.

We formally prove that a verifier that is theoretically sound (i.e., bounds the full-precision output correctly) is not necessarily practically sound, that is, it might not bound the actual output correctly in a given deployment environment.

The problem has practical implications. Recently, (Shanmugavelu et al., 2025) demonstrated that adversarial inputs can be generated simply by permuting the order of associative operations in the deployment environment.

We demonstrate a more severe, practically exploitable vulnerability as well: the deployed network is shown to be *fundamentally different* from the theoretical network because the behavior of the network can be changed arbitrarily with *deployment-sensitive backdoors*.

Thus, *the deployment environment should be a fundamental part of any verification effort* because otherwise an attacker can hide potentially harmful behaviors from the verifier if enough information about the deployment environment is available.

We summarize our contributions below:

- We prove that verifiers that are theoretically sound are not necessarily sound for deployed networks

- We demonstrate that deployed networks may differ arbitrarily from the full-precision model, with the help of backdoors triggered by features of the environment

---

[1]University of Szeged, Szeged, Hungary [2]HUN-REN–SZTE Research Group on Artificial Intelligence, Szeged, Hungary. Correspondence to: Attila Szász <szasz@inf.u-szeged.hu>.

*Proceedings of the 42nd International Conference on Machine Learning*, Vancouver, Canada. PMLR 267, 2025. Copyright 2025 by the author(s).

- We complement our theoretical analysis by an empirical evaluation, showing that the state-of-the-art verifiers that are theoretically sound indeed fail to correctly bound the deployments of our backdoored networks[1]

## 2. Related work

Verification methods can be classified in various ways (Li et al., 2023; Albarghouthi, 2021; Huang et al., 2020; Liu et al., 2021), but from a safety perspective, the most important classification categorizes algorithms based on their completeness and soundness. A verification algorithm is *sound* if every property it predicts to be true is true, and *complete* if it predicts every true property to be true. (Different terminology has also been used in the literature (Bunel et al., 2020)).

Most proposed methods assume a special class of neural networks, typically ReLU networks. Sound (but not necessarily complete) methods, such as (Singh et al., 2018; 2019a; Zhang et al., 2018; Xu et al., 2021), are often based on bound propagation (Xu et al., 2020) and linearization. These methods deal with floating point computations as well and bound the full-precision value correctly.

Methods claimed to be both sound and complete often formalize the verification problem as a satisfiability modulo theories (SMT) problem (Katz et al., 2017; Ehlers, 2017) or a mixed-integer linear programming (MILP) model (Dutta et al., 2018; Tjeng et al., 2017) and use SMT or MILP solvers to solve them.

The main disadvantages of these methods include being NP-complete (Katz et al., 2017), and relying on sophisticated solvers that might include heuristics that break soundness (Zombori et al., 2021). To mitigate the scaling issue, some methods (Tjeng et al., 2017; Singh et al., 2019b) incorporate sound (but not complete) approaches in a preprocessing phase to make the SMT or MILP models more manageable by reducing the number of integer variables.

In addition, a branch-and-bound (BaB) verification framework was also proposed for sound and complete verification (Bunel et al., 2020). The basic idea is to recursively divide the verification problem into simpler subproblems, where inexpensive sound methods can prove robustness. Verifiers based on this framework mainly differ in how they solve the subproblems and how they perform the branching process. The main advantage of the BaB framework is that the verification process can be performed partially (Xu et al., 2021) or fully (Wang et al., 2021; Palma et al., 2021; Ferrari et al., 2022) on a GPU, significantly increasing the efficiency of the verification.

---

[1]Source code: https://github.com/szasza1/no_soundness

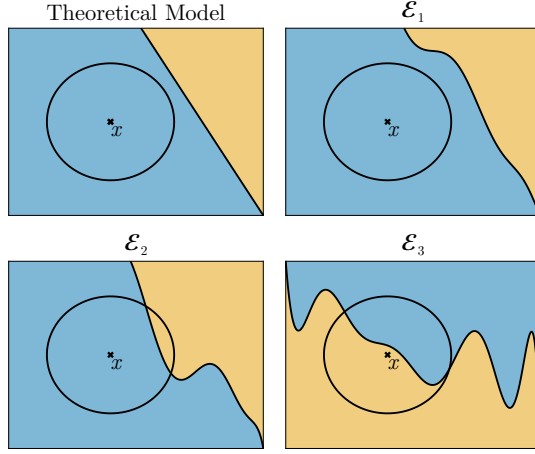

Figure 1. A malicious binary classifier with a decision boundary and two classes in two colors (conceptual illustration). The theoretical network and its three different deployments are illustrated, including a deployment in an adversarial environment ($\mathcal{E}_3$) where a malicious behavior (flipping classes) manifests itself. An input $x$ and its sensitivity domain are also shown.

The work of (Zombori et al., 2021) also addresses numeric vulnerabilities in verification, but they exploit the numerical issues of the verification algorithm itself (specifically, the MILP solver). These kinds of exploitable issues are not present in sound verifiers based on bound propagation. Instead, here, we focus on exploiting the discrepancy between full-precision models and their deployment that provably mislead every sound verifier we are aware of.

(Jia & Rinard, 2021) also address related issues. Their work can be considered an adversarial attack on verification and not a generic and provable framework we propose. It works only if the deployment and the verification algorithms use a different numerical precision, unlike our constructions.

While we focus on the problems stemming from floating point arithmetic, fixed-point arithmetic should be mentioned as an interesting alternative path where quantization can be implemented with a bounded rounding error (Lohar et al., 2023), allowing for practically sound verification.

## 3. An Intuitive Overview

Here, we discuss the main intuition behind our work in an informal manner, as well as some of the misunderstandings we often encounter.

**Main idea.** Neural networks are functions that have real valued parameters. This real valued version is the *theoretical* model. A *deployed* neural network model, however, is almost always computed using floating point arithmetic, furthermore, deployments are inherently stochastic due to the non-associative nature of floating point arithmetic and

parallelization. Our main message is that—while current verification efforts always target the theoretical model—we should target the deployed model. The main reason is that a deployed model is a fundamentally different mathematical object and this difference allows an attacker to design deployment-sensitive behaviors that remain hidden when verifying the theoretical model of the same network without taking the deployment environment into account.

**What this work is not.** Our work is *not* about the floating point issues related to the verification of the theoretical model of the neural network. For example, doing the verification in full precision arithmetic *does not solve* the problem we raise here because that way, one verifies the theoretical model and not the deployed network, except if full precision is used in deployment as well. Also, floating point arithmetic is a problem here *not* because of rounding errors, but because of its non-associativity and the stochastic nature of the computation depending on the deployment environment.

**An illustration.** Figure 1 is a conceptual illustration of a binary classifier that was manipulated by an attacker in order to produce malicious behavior in a specific deployment environment that we call adversarial environment. In this illustration, the theoretical model of the network does not show malicious behavior and it is *safe* for input $x$ as well, that is, the sensitivity domain of $x$ is inside the class of $x$. In non-adversarial deployments (environments $\mathcal{E}_1$ and $\mathcal{E}_2$) the deployed model shows variations relative to the theoretical model and it might or might not be safe for $x$. However, in the adversarial deployment environment $\mathcal{E}_3$, the model is unsafe for most inputs by design, as the class predictions are flipped. Note that this behavior is not triggered by specific inputs, instead, it is *triggered by the deployment environment*. We discuss such malicious networks in Section 7.

# 4. Background and Notations

Here, we introduce the basic notations our study focuses on: the neural network and the verification problem. We also discuss some basic properties of floating point computation briefly.

## 4.1. Theoretical Model of Neural Networks

Our formulation is inspired by (Ferrari et al., 2022). Let the *theoretical neural network* be the function $f(.;\theta): \mathbb{R}^n \mapsto \mathbb{R}^m$, where $\theta \in \mathbb{R}^k$ is a vector of constant real parameters. Throughout the paper, we work with classifier networks, that is, the network $f(.;\theta)$ classifies each input $x \in \mathbb{R}^n$ to one of $m$ classes. The class label of $x$ is given by $y(x) = \arg\max_i f(x;\theta)_i$, that is, the index of the maximal output value. Note that in many applications the input is often restricted to a subset of $\mathbb{R}^n$. Here, we allow every input,

without loss of generality.

## 4.2. Verification Problem

The robustness verification problem seeks to decide whether every input in some small neighborhood of a fixed input $x^*$ has the same class label as $x^*$. More formally, we are given an input domain $D \subseteq \mathbb{R}^n$ that defines the small environment. $D$ is usually defined by a norm $p$ and a parameter $\epsilon$: $D_{\epsilon,p}(x^*) = \{x: \|x - x^*\|_p \leq \epsilon\}$.

We wish to prove that over this domain, every input satisfies a safety property $P$. In our case, $P \subseteq \mathbb{R}^n$ captures the fact that the input has the same label as $x^*$. Let us assume the class label of $x^*$ is $y(x^*)$. Then, $P(x^*) = \{x: f(x;\theta)_{y(x^*)} = \max_i f(x;\theta)_i\}$. The goal of verification is to prove that the property $P$ holds over the input domain, that is, $D_{\epsilon,p}(x^*) \subseteq P(x^*)$.

To make our equations simpler, but without loss of generality, we extend the neural network with an additional affine layer that computes an output vector of dimension $m$ by computing $f(x;\theta)_{y(x^*)} - f(x;\theta)_i$ for every index $i$. This assumption simplifies the formulation of the verification problem, which now amounts to proving

$$\forall x \in D_{(\epsilon,p)}(x^*), \ f(x;\theta) \geq 0, \tag{1}$$

where, with a slight abuse of notation, we used the same $f$ to denote the modified network. From now on, $f$ will denote this modified function.

## 4.3. Classification of Verifiers

A verifier is an algorithm to prove the property in Equation (1) for a given $x^*$. The potential outputs of a verifier include *true, false* and *unknown*. The verifier is called *complete* if for every $x^*$, for which Equation (1) is true, it returns true. The verifier is called *sound* if for every $x^*$, for which it returns true, Equation (1) is true.

Sound verifiers have been proposed for special classes of neural networks (see Section 2). However, they are sound only in the sense of bounding $f(.;\theta)$ (the full-precision model) from below, which—as we argue in this paper—is not the right target for verification.

## 4.4. Floating-Point Issues

A floating-point number is represented as $s \cdot b^e$, where $s$ is the signed significand, $b$ is the base (usually, $b = 2$), and $e$ is the exponent. Available floating-point implementations mainly differ in the number of bits used to represent the significand and the exponent. The IEEE 754-1985 standard introduced the well-known double and single precision formats. In double-precision (or *binary64*) representation, the exponent and the significand are represented by 11 and

52 bits, respectively. In single-precision (or *binary32*), the exponent is represented by 8 bits, and the significand by 23 bits. In both cases, 1 bit is used to store the sign.

Since floating-point arithmetic has finite precision, rounding is applied to determine the floating-point result of mathematical expressions, making the arithmetic *order-dependent* (or *non-associative*). For the same expression, different rounding modes and operation orders can introduce numerical errors of varying magnitudes. Next, we demonstrate the two most significant issues that arise with floating-point arithmetic through examples.

**Non-associative operations.** Different orders of operations in a summation can lead to different results. For example $2^{53} + 1 - 2^{53} = 0$ when computed in double-precision arithmetic and with rounding towards $-\infty$ (in this fixed order). However, if we change the order to $2^{53} - 2^{53} + 1$, the result will be 1 in the same environment.

**Precision.** Different floating-point representations (such as double and single precision) have a different number of bits to represent the significand. Thus, for example, the result of $2^{24} + 1 - 2^{24}$ is 0 under single precision, but 1 under double precision.

## 5. Deployed Verification

The effect of different deployment environments have already been investigated in (Schlögl et al., 2023), although not from the point of view of verification. It was shown that different platforms—that differ in number representation, parallelization, hardware, and optimizations during convolution—significantly impact the output of deployed neural networks.

Here, we define the *deployed verification problem* and show that it differs significantly from the theoretical problem in Equation (1).

### 5.1. Deployed Neural Networks

For a theoretical model $f(.;\theta)$ and a deployment environment $\mathcal{E}$, let $r(.;\theta,\mathcal{E}) : X \mapsto Y$ be the deployed network. The notation $\mathcal{E}$ fully captures every property of the environment, including every detail of number representation, hardware, software optimizations, operation order, and potentially the stochasticity of the environment.

The idea is that $r(.;\theta,\mathcal{E})$ *implements* $f(.;\theta)$ *in* $\mathcal{E}$. This means that $r(.;\theta,\mathcal{E})$ computes the same function as $f(.;\theta)$, and the (theoretically) associative operations are executed in some (potentially random) order defined by $\mathcal{E}$, and the number representation and rounding details are also defined by $\mathcal{E}$.

Let us now elaborate on $r(.;\theta,\mathcal{E})$ and discuss a few observations about the relationship of $f(.;\theta)$ and $r(.;\theta,\mathcal{E})$.

**Domain and range.** The domain $X$ is restricted, more precisely, $X \subset \mathbb{R}^n$, where $X$ is defined by the representable numbers in $\mathbb{R}^n$ within $\mathcal{E}$. The range $Y$ behaves similarly (assuming $\mathcal{E}$ is deterministic), that is, $Y \subset \mathbb{R}^m$, where $Y$ is defined by the representable numbers in $\mathbb{R}^m$ within $\mathcal{E}$.

**Range in stochastic environments.** The environment can be stochastic, depending on a number of factors such as randomly changing operation order, or even unpredictable rounding due to using several different platforms in parallel. In this case, we do not model the distribution of the output. Instead, we define the range to be the representable subsets of $\mathbb{R}^m$, and the value of $r(x;\theta,\mathcal{E})$ is the subset containing the vectors with a probability larger than zero. In other words, $Y \subset 2^{\mathbb{R}^m}$, defined by the representable numbers in $2^{\mathbb{R}^m}$ within $\mathcal{E}$.

**Different Output.** Even for a representable input $x \in X$, and a deterministic $\mathcal{E}$, we have $r(x;\theta,\mathcal{E}) \neq f(x;\theta)$ in general. First of all, $\theta$ might not be representable exactly. Even if $\theta$ is representable, due to floating point issues, the output of the deployed network will typically be different, except in rare cases when all the sub-computations are exactly representable. In Section 7 we argue that these differences can be *arbitrary*.

### 5.2. Verification of Deployed Networks

Given an input $x^* \in X$, the verification problem is now to prove that

$$\forall x \in D_{\epsilon,p,\mathcal{E}}(x^*)\, \forall z \in r(x;\theta,\mathcal{E}),\ z \geq 0, \qquad (2)$$

where $D_{p,\epsilon,\mathcal{E}}$ is defined below. The problem is formulated assuming a stochastic environment, but note that the deterministic environment is a special case of the stochastic one. Apart from this, and the differences between $f$ and $r$ discussed above, there are other significant differences from Equation (1) regarding $D_{\epsilon,p,\mathcal{E}}(x^*)$ and the underlying property $P_{\mathcal{E}}(x^*)$.

**Verified domain.** For an $x^* \in X$, the verification problem will have a domain $D_{p,\epsilon,\mathcal{E}} = \{x \in X : \|x - x^*\|_p \leq \epsilon\}$. Now, it is unclear whether $D_{p,\epsilon,\mathcal{E}} = D_{p,\epsilon} \cap X$ because the rounding issues affecting $D_{p,\epsilon,\mathcal{E}}$ might depend on the verification algorithm as well, but for simplicity we will use $D_{p,\epsilon,\mathcal{E}} = D_{p,\epsilon} \cap X$ as a definition of $D_{p,\epsilon,\mathcal{E}}$.

**Verified property.** The properties $P$ and $P_{\mathcal{E}}$ significantly differ because we have $P_{\mathcal{E}}(x^*) = \{x \in X : r(x;\theta,\mathcal{E})_{y(x^*)} = \max_i r(x;\theta,\mathcal{E})_i\}$. Thus, in general, $P_{\mathcal{E}} \not\subset P$ because some negative outputs of $f$ might become zero for $r$ due to rounding. So, here, we do not have $P_{\mathcal{E}} = P \cap X$. (Also, obviously, $P \not\subset P_{\mathcal{E}}$.)

# 6. No Soundness in Deployment

Clearly, *practically complete* (but not sound) solutions are known: every solution based on a heuristic search for adversarial examples in the given domain is a complete method for the deployed problem, e.g., (Kurakin et al., 2017; Madry et al., 2018), although *only if the search uses the deployed implementation of the network*. We only need the method to return true if no adversarial input is found: if there are no adversarial inputs, none will be found, making the method complete.

Thus, we focus on soundness. Here, we examine *theoretically sound* methods from the literature (see Table 3 for a list) and investigate whether they are *practically sound*, that is, whether they are sound regarding the deployed verification problem in Equation (2). We will show that *in general the answer is no*. In particular, we are not aware of any method that is sound in stochastic environments, despite explicit claims (Singh et al., 2025).

In the following, we first discuss preliminary assumptions and then we address techniques applied in the literature to achieve soundness, indicated in Table 3: interval bound propagation (IBP) (Gowal et al., 2018) that is based on interval arithmetic (Alefeld & Herzberger, 1983), and more advanced propagation methods based on symbolic approaches such as affine arithmetic (de Figueiredo & Stolfi, 2004).

## 6.1. Preliminaries

In our theoretical discussion, we shall focus on a very simple class of functions: the sum of the input variables. The input variables will be zero-length intervals, that is, constants that are representable in the deployment environment. Since the problems we identify can be demonstrated already in this simple function class with constant inputs, generalizations of this class, such as neural network architectures, will inherit these problems.

The sum computation is associative, furthermore, it can be parenthesized arbitrarily, leading to a large variety of possible binary expression trees. Here, we will assume that the environment $\mathcal{E}$ defines two aspects: the floating point representation (along with the rounding mode) and the set of expression trees with non-zero probability. Thus, here, an *expression tree* defines a deterministic hierarchy of binary sum operations, all of which are computed using the same precision and rounding mode, leading to a deterministic output for any input given in the leaves.

Since the sum function does not use any parameters, we will omit $\theta$ from our notation. Thus, here, $f(x) = \sum_i x_i$ and $r(x; \mathcal{E})$ is the deployed version of the sum using the number representation and expression tree set defined by $\mathcal{E}$. Let us introduce the notation $r(x; \mathcal{E}) = \{r(x; \mathcal{E}, o_i) : o_i$ is a possible expression tree in $\mathcal{E}\}$, and let $L_r$ and $U_r$ be the

minimum and maximum elements of $r(x; \mathcal{E})$, respectively.

The proofs of the propositions are given in Appendix A.

## 6.2. Interval Bound Propagation

When using interval arithmetic on a given expression tree, each binary addition will have two intervals as inputs and will output an interval that also considers the numerical error due to rounding: if the lower or upper bound is not representable, it is rounded towards $-\infty$ or $\infty$, respectively.

Our first proposition will state a nice soundness property of IBP in two particular cases: (1) when we wish to bound the full-precision value, and (2) when we wish to bound the floating point value and we also know a specific expression tree that minimizes or maximizes this value.

**Proposition 6.1.** *Let $[a, b]_o = f([x, x])$ ($x \in X$) be the interval evaluation of $f$ using expression tree $o$ and the floating point representation defined by $\mathcal{E}$. Then, (1) $f(x) \in [a, b]_o$, (2) if $r(x; \mathcal{E}, o)$ minimizes or maximizes $r(x; \mathcal{E})$ then $a \leq L_r$ or $U_r \leq b$, respectively.*

In the special case of a deterministic environment, this means that if we know the deterministic expression tree and compute IBP accordingly, then IBP will be sound, since this expression tree will both minimize and maximize the floating point output. Unfortunately, if we do not know the deterministic order, or if the environment is stochastic, then *there is no guarantee that IBP will be practically sound*, as formulated by the next proposition.

**Proposition 6.2.** *For any environment $\mathcal{E}$ that allows for every correct expression tree and uses IEEE 754 floating point representation with any fixed rounding mode, there is an expression tree $o$, and input $x \in X$, such that for the interval evaluation $[a, b]_o = f([x, x])$ in environment $\mathcal{E}$ we have $L_r < a$ or $b < U_r$.*

Recall, that in practice, deployment environments are often inherently stochastic (Schlögl et al., 2023). Proposition 6.2 means that in stochastic environments *we must find an expression tree that guarantees practical soundness*, because not all of them do in general. This task depends on the environment but, for example, the expression tree that minimizes the output of $r(.; \mathcal{E})$ is a suitable choice for bounding the output from below in any environment according to Proposition 6.1.

## 6.3. Notes on Computational Complexity

Practically sound verification with IBP, thus, requires finding special expression trees. One might think that some orderings—such as adding numbers in a decreasing order, or maybe in a decreasing order according to absolute value—might minimize $r(.; \mathcal{E})$ and thus might be suitable to make IBP sound. This is not the case.

**Proposition 6.3.** *For any environment $\mathcal{E}$ that allows for every correct expression tree and uses IEEE 754 floating point representation with any fixed rounding mode, there is an input $x \in X$, such that we have $L_r < r(x; \mathcal{E}, o)$ for $o \in \{$decreasing-order, decreasing-absolute-value-order$\}$.*

Note that good approximation algorithms might exist for the minimum and maximum output, but in a verification mindset, we wish to eliminate any mistakes completely, otherwise the system remains vulnerable to attacks.

We hypothesize that finding the expression tree that maximizes or minimizes the output is an NP-hard problem in general. It was shown in (Kao & Wang, 2000) that a very similar problem, namely finding the order that minimizes the worst-case error is NP-hard in the general case when both positive and negative numbers are added.

### 6.4. Symbolic Bound Propagation

Due to the so called dependency problem, naive interval arithmetic might greatly overestimate the output interval of multivariate functions of a large complexity. Symbolic approaches mitigate this problem via propagating symbolic representations instead of values. We briefly summarize two important approaches here, for more details and proofs please refer to Appendix A.1.

Our main observation is that, in our special case of computing the sum of zero-length intervals, these symbolic methods simplify to non-symbolic interval methods.

**Polyhedra-based approaches.** In this special case, polyhedra-based approaches like DeepPoly (Singh et al., 2019a) and CROWN (Zhang et al., 2018; Xu et al., 2021; Wang et al., 2021) first find the symbolic expression for the sum of the input variables. During sound verification, however, the implementations evaluate this formula using interval arithmetic, based on some binary expression tree defined by the verifier. Thus, the method is equivalent to IBP discussed in Section 6.2. In other words, this method is not practically sound either in the general case.

**Zonotope-based approaches.** DeepZ (Singh et al., 2018) and RefineZono (Singh et al., 2019b) are based on the zonotope abstraction, using affine arithmetics (de Figueiredo & Stolfi, 2004) to propagate bounds throughout the network, computing the so-called zonotope domain (Ghorbal et al., 2009). This case, too, simplifies to interval analysis in our special case, only the intervals will be strictly wider than those computed by IBP due to the application of a technique proposed in (Miné, 2004).

Since this latter case is not equivalent to IBP, we explicitly state propositions describing the zonotope domain, indicating that the verification is not practically sound in general, very similarly to IBP.

**Proposition 6.4.** *Proposition 6.1 holds also when computing the interval using the widening technique in (Miné, 2004) used by (Singh et al., 2019b).*

**Proposition 6.5.** *Proposition 6.2 holds also when computing the interval using the widening technique in (Miné, 2004) used by (Singh et al., 2019b).*

## 7. Practical Attacks on Verifiers

Here, based on our the theoretical results, we design neural networks with the purpose of fooling state-of-the-art sound verifiers in practical deployment. The basic idea is that we design small detector neurons that are triggered by certain properties of an environment—such as precision or the order of computation—in order to activate arbitrary adversarial behavior.

**Adversarial networks.** These detector neurons can be inserted into any neural network using the method proposed in (Zombori et al., 2021). This way, they can enable backdoors that can alter the behavior of the network arbitrarily, triggered by a specific property of the environment. In Appendix B we discuss the technical details of creating the complete backdoored networks. Here, we detail our detector neurons.

**The detector neuron.** The generic scheme of a detector neuron is the following. We use a linear neuron with $n$ inputs without an activation function. We assume that the neuron gets the constant input vector of all ones $(1, \ldots, 1)$. This way, the output of the neuron is $w_1 + \cdots + w_n + b$, that is, we sum the edge weights and the bias. We will call this order of summands without parenthesis the *default expression tree*, often implemented in environments using a single-threaded CPU.

Note that assuming a constant input for the detector neurons is not essential because, on the one hand, our constructions can easily be generalized to a random input point from a bounded interval, and on the other hand, even a constant input can be fabricated when embedding the neuron into the host network (see Appendix B).

**Preliminaries.** We assume that the deployment environment uses IEEE 754 floating point representation with the default round-to-nearest rounding. We will use a special number $\omega$, which is defined as the smallest representable positive number such that the next representable number after $\omega$ is $\omega + 2$. For example, for the binary32 format $\omega = 2^{24}$, and for the binary64 format $\omega = 2^{53}$. Using the default rounding mode, $\omega + 1 = \omega$.

### 7.1. Detecting Precision

We define the detector neuron as $\omega + 1 - \omega$. Assuming the default expression tree (summing left to right) this detector

neuron will output 0 if the precision of the deployment is the same as the precision represented by $\omega$, and it will return 1 in higher precision deployments.

For a verifier that uses a given precision during verification, we will integrate this detector neuron into the adversarial network in such a way that the adversarial behavior will be triggered by a precision different from the one used by the verifier. This way, we can test whether the verifier is able to cover every precision that is possible in deployment.

## 7.2. Detecting Expression Trees

For detecting expression trees, we define three different detector neurons. The idea behind these definitions is that the three detectors represent an increasingly challenging problem for verifiers.

In each case, these detectors will be integrated into adversarial networks so that using the default expression tree activates the normal behavior of the network. To be more precise, if evaluating the detector in the default order results in zero then the adversarial behavior is triggered by any non-zero output of the detector, and if the default order results in a non-zero output then the output of zero triggers the adversarial behavior.

### 7.2.1. A VERIFIER-FRIENDLY DETECTOR

The detector neuron returns the following sum of $(2h_1 + 1)h_2$ summands:

$$\underbrace{\underbrace{\frac{\omega}{h_1} + \cdots + \frac{\omega}{h_1}}_{h_1\times} + 1 + \underbrace{\frac{-\omega}{h_1} + \cdots + \frac{-\omega}{h_1}}_{h_1\times} + \cdots,}_{h_2\times} \quad (3)$$

where the bias weight is zero and thus omitted.

Assuming the default expression tree the value of this sum is 0 in deployment, thus, the value 0 will trigger normal behavior.

For verifiers, dealing with this sum is relatively easy, because even simple interval arithmetic will be sound, covering every possible order, if the analysis is executed according to the default expression tree. At the same time, this trigger will be activated often, because there is a large number of possible orderings that return a value from $[1, h_2]$. Note that the full-precision value is $h_2$. In our experiments we will use $h_1 = 4$ and $h_2 = 15$.

### 7.2.2. VERIFIER-UNFRIENDLY DETECTORS

Our first unfriendly detector returns the sum of $h + 2$ summands

$$\underbrace{\frac{2}{h} + \cdots + \frac{2}{h}}_{h\times} + \omega - \omega \quad (4)$$

Assuming the default expression tree the value of this sum is 2 in deployment, thus, any non-zero value will trigger the normal behavior, and the value of zero triggers adversarial behavior.

It is a harder expression for verifiers, because now, a verification using interval arithmetic on the default expression tree will no longer be sound: it will not contain the output 0, which is a possible output. Indeed, if we sum the edge weights in an order where $\omega$ is in a position earlier than position $h/2$ and we add the bias $-\omega$ in the end, the result is 0. But, in this case, there is a large number of orders that do result in covering 0 (and thus a practically sound behavior). In fact, the only order of the edge weights that does not result in covering 0 is the default order.

For verification, the hardest problem we will test is the sum

$$\underbrace{1 + \cdots + 1}_{h\times} + \omega - \omega, \quad (5)$$

which has much fewer possible orderings of the edge weights that result in covering the output 0 using interval arithmetic. In fact, we cover zero only if $\omega$ is first or second in the order of summation (assuming the bias is added last). This means that verifiers that are not guaranteed to cover all possible orders will almost certainly fail to detect the adversarial behavior.

As for the parameter $h$, we set $h = 512$ for both unfriendly detectors.

## 8. Empirical Evaluation

To demonstrate the practical relevance of our observations, we evaluated state-of-the-art verifiers using our adversarial networks to test whether they indeed cover all possible executions in a number of different deployment environments. As predicted, the answer we found is *no*.

### 8.1. Adversarial Networks

In our evaluation, we used an MNIST network checkpoint made available by (Wong & Kolter, 2018). This checkpoint was used to evaluate MIPVerify (Tjeng et al., 2017) as well as the attacks proposed by (Zombori et al., 2021). This network has two convolutional layers with stride 2: one with 16 and one with 32 filters of size 4×4, followed by a 100 neuron fully connected layer. ReLU activations are used by all the neurons. The network was trained to be robust against attacks within a radius of 0.1 in the $l_\infty$-norm (Wong & Kolter, 2018).

We augmented this network with our detectors presented in Section 7 using the methodology proposed in (Zombori et al., 2021), as detailed in Appendix B. We created five adversarial networks: two precision-based attacks (one with

*Table 1.* Test accuracy of backdoored networks.

| Attack | Precision of the environment | |
|---|---|---|
| | 32-bit | 64-bit |
| Precision, 32-bit adversarial | **0.11%** | 98.11% |
| Precision, 64-bit adversarial | 98.11% | **0.11%** |

*Table 2.* Test accuracy of backdoored networks, evaluated with different batch size and environment settings.

| Attack | Batch Size | PyTorch | | Flux | |
|---|---|---|---|---|---|
| | | CPU | GPU | CPU | GPU |
| Order1 | 1 | **0.11%** | **0.11%** | 98.11% | **0.11%** |
| Order2 | 1 | 98.11% | 98.11% | 98.11% | **0.11%** |
| Order3 | 1 | 98.11% | 98.11% | 98.11% | 98.11% |
| Order1 | 10 | **0.11%** | 98.11% | **0.11%** | 98.11% |
| Order2 | 10 | 98.11% | 98.11% | 98.11% | 98.11% |
| Order3 | 10 | 98.11% | 98.11% | 98.11% | 98.11% |

adversarial behavior in 32-bit environments, and one in 64-bit environments) based on the detector in Section 7.1, and three expression tree based attacks described in Section 7.2 based on the detectors in Equations (3) to (5). We will refer to these three attacks as Order1, Order2, and Order3, respectively.

## 8.2. Effectiveness of our Adversarial Networks

Here, we wish to demonstrate that the adversarial behavior of our networks is indeed activated in certain environments.

### 8.2.1. PRECISION ATTACKS

In Table 1, the accuracy of the adversarial networks with precision attacks is presented. The first row corresponds to the configuration where the backdoor changes the prediction under 32-bit precision, while the second row corresponds the 64-bit precision trigger. The columns correspond to 32-bit and 64-bit inference. It is clearly visible that our backdoors indeed trigger adversarial behavior as designed. Note that the adversarial behavior has larger than zero accuracy due to some of the test examples that are misclassified by the host model.

### 8.2.2. EXPRESSION TREE ATTACKS

**The environments.** The effective expression tree in deployment depends on numerous factors, such as hardware specifications, memory size, underlying linear algebra libraries, machine learning frameworks, optimizations, batch size during inference, and so on (Schlögl et al., 2023). We defined 8 different environments that provide a sufficient diversity to support our claims. The libraries we tested were PyTorch and Flux on Julia. As for hardware, we tested CPU and GPU, and we also tested different batch sizes during inference. For a more detailed description please refer to

Appendix C.1.

**Results.** Table 2 contains the accuracy results over different environments. Three of our backdoors are activated in at least one environment, which gives further support that these environments execute different expression trees, including those that we used as backdoors. The Order3 adversarial network is an exception because—as we explained before—it has very few expression trees that trigger its adversarial behavior. Nevertheless, this does not make the Order3 network safe, and this fact should still be detected by sound verifiers.

## 8.3. Pool of Verifiers

Table 3 lists the verifiers that we tested. Here, we briefly describe them, for more detail, please refer to Appendix C.2.

MIPVerify, RefineZono, and RefinePoly are claimed to be sound and complete verifiers. They rely on a cheap sound bounding method that is then refined using mixed-integer linear programming (MILP) to achieve completeness.

RefineZono and RefinePoly rely on DeepZ and DeepPoly as their sound bounding method, respectively. DeepZ and DeepPoly are both claimed to be sound under floating-point arithmetic.

$\beta$-CROWN BaB is built on the fast and sound (but incomplete) bound propagation algorithm $\beta$-CROWN and integrates it with a branch-and-bound framework to ensure completeness. The verification process can be configured to run on either CPU or GPU and supports numerical representations in both 32-bit and 64-bit precision. GCP-Crown is also a BaB method similar to $\beta$-CROWN, but it is capable of managing general cutting-plane constraints.

## 8.4. Attacking Verifiers

Here, we verify our adversarial networks with our pool of verifiers. Note that here, the verifiers do not consider the deployment environments explicitly, instead, the verifiers themselves have their own deployment environments, in which they are executed as indicated in Table 3 (see Appendix C.2 for a detailed discussion).

As for the deployment environments of our networks, we know from Section 8.2.2 that for all the adversarial networks there is at least one environment where the adversarial behavior is observed, except for Order3. Thus, *it is a rightful expectation* that the verifiers find the planted backdoor.

In the case of Order3, we argue that we should also require the verifiers to find the backdoor, because in general it is very hard to explicitly model the set of possible expression trees in a given complex environment, so practically sound verifiers should be prepared for all of them.

*Table 3.* The vulnerability of the verifiers to our four attacks and to the attack of (Zombori et al., 2021). The Precision attack is adversarial to the 32-bit environment, except for 32-bit $\beta$-CROWN, where it is adversarial to 64-bit.

| Verifier | Ver. Env. | Bounding | Precision | Order1 | Order2 | Order3 | (Zombori et al., 2021) |
|---|---|---|---|---|---|---|---|
| MIPVerify (Tjeng et al., 2017) | 64-bit, CPU | IBP | **unsound** | sound | **unsound** | **unsound** | **unsound** |
| MN-BAB (Ferrari et al., 2022) | 64-bit, GPU | Polyhedra | **unsound** | sound | **unsound** | **unsound** | **unsound** |
| $\beta$-CROWN BaB (Wang et al., 2021) | 32-bit, CPU | Polyhedra | **unsound** | sound | **unsound** | **unsound** | [no 32-bit model] |
| $\beta$-CROWN BaB (Wang et al., 2021) | 64-bit, CPU | Polyhedra | **unsound** | sound | **unsound** | **unsound** | sound |
| $\beta$-CROWN BaB (Wang et al., 2021) | 64-bit, GPU | Polyhedra | **unsound** | sound | **unsound** | **unsound** | sound |
| GCP-CROWN (Zhang et al., 2022) | 64-bit, CPU | Polyhedra | **unsound** | sound | **unsound** | **unsound** | sound |
| DeepPoly (Singh et al., 2019a) | 64-bit, CPU | Polyhedra | **unsound** | sound | sound | **unsound** | sound |
| RefinePoly (Singh et al., 2019a) | 64-bit, CPU | Polyhedra | **unsound** | sound | sound | **unsound** | **unsound** |
| DeepZ (Singh et al., 2018) | 64-bit, CPU | Zonotope | **unsound** | sound | sound | **unsound** | sound |
| RefineZono (Singh et al., 2019b) | 64-bit, CPU | Zonotope | **unsound** | sound | sound | **unsound** | [Gurobi error] |

Table 3 contains the results of the verification of the first 100 examples from the MNIST test set. The label *sound* indicates that the verifier was able to detect the backdoor each time, returning a 0% verified robust accuracy. Otherwise the result is labeled *unsound*.

### 8.5. Discussion

**No practically sound verifiers.** The results in Table 3 reveal that none of the listed approaches are practically sound, given that they missed the backdoor in the most difficult Order3 network as well as in the Precision network.

**Difficulty matters.** It is also clear that as the task becomes more difficult, more and more verifiers fail. The Order1 network is solved by all the verifiers since bound propagation in the default operation order covers the adversarial output as well. The relatively easy Order2 network is solved by some of the verifiers, which could be due to the wider intervals used during propagation, and potentially the operation order used by the verifier could deviate from the default, thereby accidentally covering the adversarial output. However, we know that these successful cases are due to lucky accidents, as we have proven theoretically that there is no guarantee for practical soundness for these methods.

**Precision matters.** All the verifiers are vulnerable to the precision attack, which provides further empirical support for our observation that verifiers should explicitly be tailored to a specific floating point precision.

(Zombori et al., 2021) propose a network similar to Order1 in that in the default order bound propagation covers the adversarial output. This network was designed to trigger numerical errors in the verifier, so only those ones are vulnerable that use optimization in some form. Our attacks, on the other hand, exploit the difference between the behavior of the full-precision model and the deployed network, and the strongest ones (Precision and Order3) are effective against all the verifiers that are only theoretically (but not practically) sound.

## 9. Conclusions and Limitations

Our main conclusion is that verifiers that are theoretically sound (cover the full-precision output despite floating point computations) are not necessarily practically sound, in that they might not cover every possible output when the network is used in a deployment environment. We are therefore proposing to focus on the behavior of deployed neural networks as opposed to the full-precision behavior.

As for limitations, our present work does not include any proposals for implementing practically sound verifiers. Here, we focused on exposing the problem, which in itself is of value, given that every verifier we are aware of suffers from this. Nevertheless, solving the problem would be important.

We believe that practically sound verifiers will have to make strong assumptions about the deployment environment, otherwise the problem becomes prohibitively expensive. For example, deterministic environments are a great advantage, but then verification must be aware of and exploit every detail. Bound propagation will be practically sound if it follows the deterministic expression tree, provided the propagation takes into account the (deterministic) precision and rounding rules of the environment.

## Acknowledgements

This work was supported by the European Union project RRF-2.3.1-21-2022-00004 within the framework of the Artificial Intelligence National Laboratory and project TKP2021-NVA-09, implemented with the support provided by the Ministry of Culture and Innovation of Hungary from the National Research, Development and Innovation Fund, financed under the TKP2021-NVA funding scheme. We thank the support by the PIA Project, a collaboration between the University of Szeged and Continental Autonomous Mobility Hungary Ltd. with the goal of supporting students' research in the field of deep learning and autonomous driving. We thank Andras Balogh for his initial feedback and our anonymous reviewers for the relevant

pointers and suggestions.

## Impact Statement

This paper presents work with the goal of advancing the field of Machine Learning. There are many potential societal consequences of our work, none of which we feel must be specifically highlighted here.

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

# Supplementary Material

## A. Proofs of our Propositions

Here, we provide the proofs of our propositions. We use the notation $\odot_\diamond$, where $\odot \in \{+, -\}$ and $\diamond \in \{-\infty, +\infty, 0, n\}$ (where $n$ represents round-to-nearest) to denote the rounding mode applied after the binary operation $\odot$.

Recall that we have introduced the notation $r(x; \mathcal{E}) = \{r(x; \mathcal{E}, o_i) : o_i \text{ is a possible expression tree in } \mathcal{E}\}$, and $L_r$ and $U_r$ are the minimum and maximum elements of $r(x; \mathcal{E})$, respectively.

We will use a special number $\omega$, the value of which depends on the underlying IEEE 754 floating point precision. In a given precision, let $\omega$ be the smallest representable positive number such that the next representable number after $\omega$ is $\omega + 2$.

For example, for the binary32 format $\omega = 2^{24}$, and for the binary64 format $\omega = 2^{53}$. Note that the closest representable number less than $\omega$ is $\omega - 1$. Also, we will use the fact that $\omega +_n 1 = \omega$, that is, the default round-to-nearest rounding mode will round $\omega + 1$ towards $-\infty$.

The definition of $\omega$ allows us to formulate the proofs without explicit reference to a specific floating point representation. However, note that in the IEEE 754 standard the lowest resolution floating point representation is the half-precision (or binary16) format. For even lower resolution floats the proofs need slight adjustments that we avoid here for readability.

**Proposition 6.1.** Let $[a, b]_o = f([x, x])$ ($x \in X$) the interval evaluation of $f$ using expression tree $o$ and the floating point representation defined by $\mathcal{E}$. Then, (1) $f(x) \in [a, b]_o$, (2) if $r(x; \mathcal{E}, o)$ minimizes or maximizes $r(x; \mathcal{E})$ then $a \leq L_r$ or $U_r \leq b$, respectively.

*Proof of (1).* This is a well-known property of interval arithmetic, included only for completeness. □

*Proof of (2).* We show that for every expression tree, the output interval computed using interval arithmetic contains the floating point result computed according to the same expression tree. Proposition (2) follows from this observation, applying it to the expression trees that maximize or minimize $r(x; \mathcal{E})$.

This can be shown by induction according to the structure of the expression tree. The statement is true for the leaves of the tree (the inputs) because they are zero-length intervals of the form $[x, x]$ where $x$ is a representable number by assumption.

All the non-leaf nodes represent the addition operator in our case. Let the input intervals for a non-leaf node be $[x_l, x_u]$ and $[y_l, y_u]$. By induction, we know that these intervals contain the floating point output of the given sub-expression. Let these be $x$ and $y$, respectively, where we thus know that $x_l \leq x \leq x_u$ and $y_l \leq y \leq y_u$. The interval addition results in the interval $[x_l +_{-\infty} y_l, x_u +_{+\infty} y_u]$. But we know that for any rounding mode $\diamond$

$$x_l +_{-\infty} y_l \leq x_l +_\diamond y_l \leq x +_\diamond y \leq x_u +_\diamond y_u \leq x_u +_{+\infty} y_u, \tag{6}$$

which proves that the non-leaf node's output contains the true floating point value $x +_\diamond y$. □

**Proposition 6.2** For any environment $\mathcal{E}$ that allows for every correct expression tree and uses IEEE 754 floating point representation with any fixed rounding mode, there is an expression tree $o$, and input $x \in X$, such that for the interval evaluation $[a, b]_o = f([x, x])$ in environment $\mathcal{E}$ we have $L_r < a$ or $b < U_r$.

*Proof.* We give simple examples where $L_r$ or $U_r$ is not included in the resulting interval.

Let us consider the expression tree $(1 + 1) + \omega$. We note that under any rounding mode $\diamond \in \{-\infty, 0, n, +\infty\}$ the interval result of the expression tree is $[\omega + 2, \omega + 2]$, because every sub-expression can be computed exactly.

When $\diamond \in \{-\infty, 0, n\}$, we have $L_r = \omega = \omega +_\diamond 1 +_\diamond 1$ and $U_r = \omega + 2 = 1 +_\diamond 1 +_\diamond \omega$. Since $L_r \notin [\omega + 2, \omega + 2]$, interval arithmetic fails to cover $L_r$, hence it is not sound.

When $\diamond = +\infty$, we have $L_r = \omega + 2 = 1 +_{+\infty} 1 +_{+\infty} \omega$ and $U_r = \omega + 4 = \omega +_{+\infty} 1 +_{+\infty} 1$. Since $U_r \notin [\omega + 2, \omega + 2]$, interval arithmetic fails to cover $U_r$, hence it is not sound. □

One could think that choosing the rounding mode $+\infty$ could result in a sound solution at least for the lower bound. This, however, is true only for the case of computing sums. It is an easy exercise to show that if we allow multiplication by negative numbers than no soundness can be achieved. In this paper, however, we focus on the sum for simplicity.

**Proposition 6.3** For any environment $\mathcal{E}$ that allows for every correct expression tree and uses IEEE 754 floating point representation with any fixed rounding mode, there is an input $x \in X$, such that we have $L_r < r(x; \mathcal{E}, o)$ for $o \in \{$decreasing-order, decreasing-absolute-value-order$\}$.

*Proof.* We provide simple counterexamples. We will use only positive numbers so the two orderings (decreasing, and decreasing in absolute value) are the same.

For a rounding mode in $\{+\infty, n\}$, consider the multiset of numbers $\{\omega, 1.25, 1.25, 1.25\}$. Here $L_r = \omega + 4$ (when adding the numbers in increasing order). Computing the sum in decreasing order gives $\omega + 6$, which is greater than $L_r$.

For a rounding mode $\diamond \in \{-\infty, 0\}$, consider the set of three numbers $\{\omega - 1, 3, 2\}$, where $L_r = \omega + 2 = (\omega - 1) +_\diamond 2 +_\diamond 3$. Computing the sum in decreasing order we get the exact value $\omega + 4 > L_r$. $\qquad\square$

### A.1. Symbolic Bound Propagation

Due to the so called dependency problem, naive interval arithmetic might greatly overestimate the output interval of multivariate functions of a large complexity. Symbolic approaches mitigate this problem via propagating expressions of input variables instead of values.

Polyhedra-based approaches compute the polyhedral domain for each neuron. That is, linear upper and lower bounds are computed for each neuron, and these are propagated in a symbolic manner. Typical representatives of these approaches are DeepPoly (Singh et al., 2019a), and CROWN (Zhang et al., 2018; Xu et al., 2021; Wang et al., 2021).

In our special case, where the function under consideration is the sum function with representable constant inputs, this approach *falls back to plain interval arithmetics*. The reason is that the summands are independent so symbolic propagation simply results in the formula that sums the input variables. During sound verification, implementations evaluate this formula using interval arithmetics, based on some binary expression tree defined by the verifier.

Thus, in the special case of computing a sum, the properties of this method are exactly the same as those of IBP in Section 6.2. In other words, this method is not practically sound either in the general case.

### A.2. Affine Arithmetics

Zonotope-based approaches are similar to polyhedra-based methods but they use affine arithmetics (de Figueiredo & Stolfi, 2004) to propagate bounds throughout the network, computing the so-called zonotope domain (Ghorbal et al., 2009). Well-known algorithms such as DeepZ (Singh et al., 2018) and RefineZono (Singh et al., 2019b) are based on this abstraction.

When the goal is soundness to floating point operations, this method is combined with a technique proposed in (Miné, 2004), which uses affine forms with interval coefficients, combined with widening the intervals with certain extra error terms defined by the floating point precision in the deployment environment $\mathcal{E}$. More precisely, when adding two intervals, the extra error terms are

$$[a_l, a_u] + [b_l, b_u] = [a_l +_{-\infty} b_l, a_u +_{+\infty} b_u] + \varepsilon([a_l, a_u]) + \varepsilon([b_l, b_u]) + m \cdot [-1, 1], \qquad (7)$$

where $m$ represents the smallest positive non-zero number representable in $\mathcal{E}$, and $\varepsilon$ is defined as

$$\varepsilon([a_l, a_u]) = \max(|a_l|, |a_u|) \cdot [-2^{-p}, 2^{-p}] \qquad (8)$$

where $p$ is the fraction size in bits for the floating-point representation. In IEEE 754, $p = 52$ for binary64 and $p = 23$ for binary32.

These extra error terms are introduced to make it possible to compute the verification in higher precision than the precision in $\mathcal{E}$, and it also allows the verifiers to use arbitrary rounding modes while staying sound in $\mathcal{E}$.

However, when applying this method to simple sum computation with constant representable inputs, the interval-coefficient affine forms will simplify to interval constants (since the inputs are independent, zero-length intervals). Thus, the behavior

will be very similar to the case when interval arithmetic is used, only the intervals will be strictly wider. The following statement captures this similarity.

**Proposition 6.4** Proposition 6.1 holds also when computing the interval using the widening technique in (Miné, 2004) used by (Singh et al., 2019b).

*Proof.* The proposition follows directly from the fact that the intervals here are strictly wider than in the case of IBP (see Equation (7)). □

**Proposition 6.5** Proposition 6.2 holds also when computing the interval using the widening technique in (Miné, 2004) used by (Singh et al., 2019b).

*Proof.* When $\diamond \in \{-\infty, 0, n\}$, we define the multiset of nine 1s and $\omega$ $\{1, 1, 1, 1, 1, 1, 1, 1, 1, \omega\}$, where $L_r = \omega$ (summing in decreasing order).

Now, let us compute the zonotope sum in increasing order using Equations (7) and (8). The zonotope result of adding the nine 1s will be an interval $[9 - \delta_l, 9 + \delta_u]$, where $0 < \delta_l \ll 1$ and $0 < \delta_u \ll 1$. The final lower bound, as a result of adding $\omega$ using Equation (7), is

$$l = (9 - \delta_l) +_{-\infty} \omega -_\diamond 2 -_\diamond 2 -_\diamond m > L_r \tag{9}$$

because $(9 - \delta_l) +_{-\infty} \omega = \omega + 8$ and $0 < m \ll 2$.

When $\diamond \in \{+\infty, n\}$, we define the multiset of nine 1.1s and $\omega$ $\{1.1, 1.1, 1.1, 1.1, 1.1, 1.1, 1.1, 1.1, 1.1, \omega\}$, where $U_r = \omega + 18$ (summing in decreasing order).

Now, let us compute the zonotope sum in increasing order using Equations (7) and (8). The zonotope result of adding the nine 1.1s will be an interval $[9.9 - \delta_l, 9.9 + \delta_u]$, where $0 < \delta_l \ll 1$ and $0 < \delta_u \ll 1$. The final upper bound, as a result of adding $\omega$ using Equation (7), is

$$u = (9.9 + \delta_u) +_{+\infty} \omega +_\diamond 2 +_\diamond 2 +_\diamond m < U_r \tag{10}$$

because $(9.9 + \delta_u) +_{+\infty} \omega = \omega + 10$ and $0 < m \ll 2$. □

Accordingly, in the case of the Zonotope-based approaches the situation is similar to the previous cases we examined: the verification is not practically sound in general, and achieving guaranteed soundness is likely to have a prohibitive computational complexity.

## B. Adding a Backdoor

We use a technique similar to that of (Zombori et al., 2021) to integrate backdoors into a convolutional architecture. A conceptual diagram of the backdoor can be seen in Figure 2.

A main difference from the construction of (Zombori et al., 2021) is that our backdoors are fully independent of the input, and are triggered purely by the environment. Note that having a constant input for the detector is not essential, we do this for simplicity only. In many applications, the input to the network is already constrained to a specific interval, and we can adapt every detector to work with constrained input (e.g., $[0, 1]$, like in image processing applications) as well.

We also introduced the $\alpha$ and $\beta$ parameters for convenience so that we can easily control whether the output of the host network is shifted by a detector output of 0 or $> 0$. Parameter configuration $\alpha_1 = 1$, $\alpha_2 = -2$, $\beta_1 = 0$, and $\beta_2 = 1$ ensures that the logits are shifted by one position when the output of the detector is greater than 0 (precisely, when the output of the detector is $\geq 0.5$, but our detector cannot output any fraction between 0 and 1). Conversely, parameter configuration $\alpha_1 = -2$, $\alpha_2 = 1$, $\beta_1 = 1$, and $\beta_2 = 0$ ensures that the logits are shifted when the detector outputs 0.

At the implementation level, for the precision-based detectors, we added one additional kernel to the 2nd convolutional layer of the host network, with weights initialized to zero and bias set to 1, thereby implementing the constant 1 input to the detector. Similarly, for the Order1, Order2, and Order3 detectors, we added 3, 15, and 15 kernels to the 2nd layer, respectively.

The patterns of the detectors are then integrated into the weight matrix of the 3rd layer constructing an additional neuron. Since this layer is already a fully connected layer, the integration is straightforward: a new column is appended to the end of

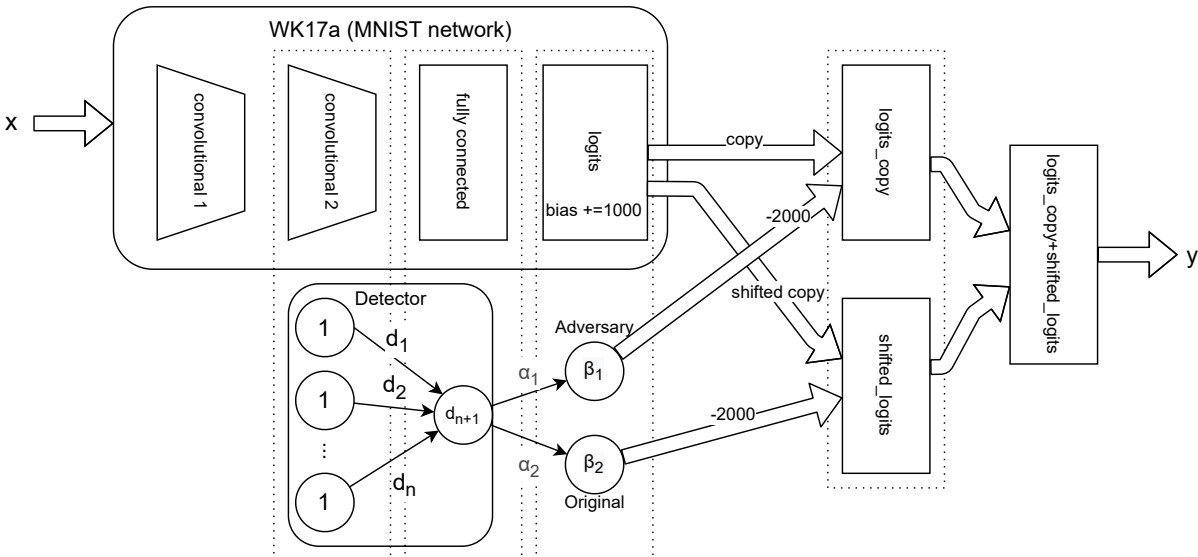

*Figure 2.* Backdoor integration. Circles represent ReLU neurons with the parameter inside the circle being the bias term. Simple arrows represent connections with the weights indicated on them.

the weight matrix (assuming a row input vector), with the last $n$ elements configured according to the detectors and the remaining elements set to zero. The bias is also configured according to the detector patterns. The shifting mechanism after the last layer of the base model is configured as described in (Zombori et al., 2021).

Since our backdoors have very large weights and are independent of the network input, they are quite obvious to detect. However, here, *our motivation was theoretical*, so we largely ignored the practical aspects of actually carrying out a similar attack. Nevertheless, we propose some ideas that could make our backdoors less detectable.

Firstly, the large weights can be distributed across several layers using a technique similar to what is described in (Zombori et al., 2021). Additionally, it is possible to make our backdoors functional only for specific input perturbations. For example, the backdoor could be designed to shift the output by one position only when the pixel value of the top-left pixel exceeds a predefined value and the numerical precision is 32-bit floating-point. A similar implementation was used in (Zombori et al., 2021).

## C. Detailed Description of our Experimental Setup

### C.1. Deployment Environments

Since our work is inherently dependent on the hardware and software specifications of the environments, we provide the most important details to ensure reproducibility. First, we detail the most important hardware specifications common to all environments, followed by a description of software-specific properties.

Hardware:

- Memory: 32 GiB

- CPU: AMD Ryzen 7 3800X 8-Core

- GPU: NVIDIA GeForce RTX 4080 SUPER

    - Driver Version: 535.183.01
    - CUDA Version: 12.2

Operating System: Ubuntu 20.04.6 LTS

PyTorch specific software versions:

- Python version: 3.12.7

- PyTorch package: 2.4.1 + cu124

Flux specific software versions:

- Julia version: 1.11.0

- Flux package: 0.14.21

- CUDA package: 5.5.2

- cuDNN package: 1.4.0

### C.2. Pool of Verifiers

Here, we detail the verifiers we tested as well as their weak points of implementation that make our backdoors functional. During the evaluation, we tested the first 100 test samples from the MNIST dataset. The robust accuracy of the base model for these samples, within a radius of 0.1 in the $l_\infty$ norm, was determined to be 97% according to all complete and sound algorithms. DeepPoly and DeepZ (that are not complete but sound) verified robust accuracies of 96% and 94%, respectively.

#### C.2.1. MIPVERIFY [2]

MIPVerify is based on the mixed-integer linear programming (MILP) formulation of the network, incorporating an effective presolve step to reduce the number of integer variables. During this step, MIPVerify aims to tighten the bounds of the problem's variables. If this step reveals that a ReLU node's input is always non-positive, it can be fixed as a constant zero. Similarly, if the input to a ReLU node is always non-negative, the ReLU can be treated as an identity function and removed from the model, reducing the number of binary variables in the formulation. MIPVerify employs a progressive approach in its presolve step. Initially, it uses fast but imprecise interval arithmetic to estimate bounds. These preliminary bounds are then refined by solving a relaxed LP problem. Finally, the MILP problem is solved to determine the precise bounds for the variables. MIPVerify is implemented in Julia, utilizing the Gurobi solver to handle LP/MILP models and the IntervalArithmetic.jl (Sanders & Benet, 2014) library for interval arithmetic operations. The bounds during interval operations are rounded outward: the lower bound is rounded down, and the upper bound is rounded up. All operations during the verification process are executed on the CPU using 64-bit precision.

MIPVerify was attacked using the 32-bit adversarial precision backdoor, leading to incorrect outputs of 97 safe answers. In the case of Order1, it correctly outputs 0% robust accuracy. However, this result arises from the fact that the environment used by MIPVerify already triggers the adversarial behavior of the networks. On the other hand, the backdoor in Order2 and Order3 remained fully invisible to MIPVerify, resulting in an incorrect verification of 97% robust accuracy.

#### C.2.2. ERAN [3]

DeepPoly, DeepZ, RefinePoly, and RefineZono are implemented within the ERAN framework. DeepPoly and DeepZ are based on polyhedra and zonotope abstract domains, respectively, and both implementations are claimed to be sound under floating-point operations. They extend their general formulation with interval coefficients and additional error terms, as explained in (Miné, 2004). The underlying linear algebra operations for DeepZ and DeepPoly are implemented in a separate C library called ELINA (Singh et al., 2025). All operations during the verification are performed on the CPU using 64-bit precision. RefinePoly and RefineZono are sound and complete algorithms that utilize the DeepPoly and DeepZ abstract domains, respectively. They initially run the DeepPoly/DeepZ analysis on the entire network, collecting bounds for all neurons. If robustness cannot be proven based on these bounds, they construct a MILP formulation of the network, similar to MIPVerify, but employ alternative heuristics to improve effectiveness.

---

[2] https://github.com/vtjeng/MIPVerify.jl
[3] https://github.com/eth-sri/eran

For the 32-bit adversarial precision backdoor, DeepPoly and DeepZ incorrectly reported robust accuracies of 96% and 94%, respectively. These results match the robust accuracy of the base model as verified by DeepPoly and DeepZ (both are incomplete algorithms). For samples not verified as robust, the algorithms output 'failed', indicating that they were unable to decide whether the input is safe or not. Both DeepPoly and DeepZ successfully detected the backdoor in Order1 and Order2, verifying 0% robust accuracy for both networks. However, for Order3, they incorrectly reported the same robust accuracy as for the base model.

Both RefinePoly and RefineZono incorrectly reported 97% robust accuracy for the 32-bit adversarial precision backdoor. However, they successfully detected the backdoor in Order1 and Order2, verifying 0% robust accuracy for these networks. For Order3, RefineZono incorrectly output 97% robust accuracy. Similarly, RefinePoly incorrectly reported 96% robust accuracy (matching DeepPoly). However, for the remaining 4 images, where the algorithm constructed a MILP model and used Gurobi to solve it, the algorithm stopped with an error.

### C.2.3. MN-BAB [4]

MN-BaB is claimed to be a sound and complete verifier. It is built upon the branch-and-bound framework with multi-neuron constraints and a dual method. Its bounding approach extends DeepPoly with optimizable parameters, as described in (Wang et al., 2021), and incorporates the HybridZonotope domain (Mirman et al., 2018). In our experiments, we executed MN-BaB on a GPU with 64-bit numerical precision. The entire algorithm, including the bounding approach and projected gradient descent, is implemented in Python using the PyTorch framework. The coefficients of the linear forms are represented as floating-point numbers rather than intervals. During arithmetic operations, only a small constant is added in certain cases to address potential numerical errors.

MN-BAB successfully reported 0% robust accuracy for Order1, but the underlying implementation already triggered the adversarial behavior, causing the network to misclassify all clean inputs. On the other hand, for all other backdoors, it incorrectly verified 97% robust accuracy.

### C.2.4. $\beta$-CROWN BAB AND GCP-CROWN [5]

$\beta$-CROWN BaB and GCP-CROWN are sound and complete algorithms implemented within the $\alpha, \beta$-CROWN framework.

$\beta$-CROWN BaB is based on the $\beta$-CROWN bounding approach, which extends $\alpha$-CROWN by introducing optimizable ($\beta$) parameters to encode split constraints. These parameters are optimized using gradient descent. The algorithm initially runs $\alpha$-CROWN to compute initial bounds for the objectives and then begins a branch-and-bound-based refinement, as described in (Wang et al., 2021), Appendix B.1. GCP-CROWN is a similar algorithm to $\beta$-CROWN BAB, but is capable of managing general cutting-plane constraints. The algorithm integrates high-quality cutting planes into the optimization objective, which is then optimized using gradient descent.

The bounding approach for both algorithms is implemented using the auto_LiRPA library (Xu et al., 2020). The coefficients of the linear forms are represented as floating-point numbers, and numerical errors are completely ignored during arithmetic operations.

$\beta$-CROWN BaB and GCP-CROWN produced similar results for all backdoors. Both algorithms successfully reported 0% robust accuracy for Order1. However, all other backdoors were undetected by these methods. $\beta$-CROWN BaB incorrectly output 97% robust accuracy. Similarly, GCP-CROWN also incorrectly reported 97 "safe" results, but it encountered errors for the remaining 3 images.

---

[4]https://github.com/eth-sri/mn-bab
[5]https://github.com/Verified-Intelligence/alpha-beta-CROWN

