# OpenReview forum: "No Soundness in the Real World: On the Challenges of the Verification of Deployed Neural Networks"
_ICML.cc/2025/Conference — ICML 2025 spotlightposter_

### Official Review · Reviewer_Twzp · 2025-03-13

**Overall Recommendation:** 5

**Summary:**

The paper provides a detailed theoretical and empirical analysis on the soundness of neural network verifiers. The paper demonstrates that a number of State of the Art verifiers can produce misleading or even wrong results when considering the evaluation of NNs on modern hardware.

## update after rebuttal
I stand with my opinion that this paper should be accepted, I appreciate the authors'  responses.

For the sake of completeness, I am copy-pasting my comment in the reviewer discussion below to make it available to the authors:

It seems to me that a key difference in opinion between reviewers lies in the question of whether the results are surprising.
I want to share my perspective on this question from two angles and would be very interested to hear your thoughts about this:

**1. Necessity of Surprise**
For some of the solvers, I personally was not surprised that they are not floating point sound.
However, despite the lack of surprise, I believe these results are timely and valuable because they empirically demonstrate a phenomenon that I suspected to be present, but have not seen evaluated to any degree as detailed as in the paper at hand.
In my opinion, we should thus not only measure this publication by the degree to which we were surprised by the outcome,
but also by the degree to which the empirical results are valuable information to the community.
And in this instance, I believe there is clearly timely, important information that warrants dissemination:
The results indicate that even **four years** after the publication by Zombori et al. NN verifiers still struggle with the same problems.
One might even argue that this paper can be seen as a "call to arms" for the NN verification community to start addressing this critical challenge.

**2. Level of Surprise**
Above, I mentioned that I was not surprised by the results for *some* solvers.
However, the authors also show that there are instances in which, e.g., the solver DeepZ is unsound w.r.t. floating point arithmetic.
On page 2 of [Singh et al. 2018] the DeepZ paper states, "Further, our transformers are sound w.r.t. floating point arithmetic.".
This is, in fact, even mentioned in the abstract. Later on in the paper, this statement is made even more precise (page 3): "To ensure soundness with respect to different rounding modes and to account for the **lack of algebraic properties such as associativity and distributivity** in the floating point world [...]". However, it is precisely this lack algebraic properties (presumably associativity) that is the reason why DeepZ produces an unsound result for Order3 in Table 3 of the paper.
In no way do I wish to attack the authors of DeepZ here, who have actually gone much further than most NN verification tools to achieve floating point soundness, but it is a prime example of just how difficult this is to achieve -- and a result which surprised me.

A possible consequence might be that we can only provide guarantees w.r.t. concrete implementations/hardware configurations; there might also be other possible solutions. But I do not see it as a burden for the authors of this paper alone to find a solution to this issue, which spans most of the NN verification community. Instead, I see this as an instance where raising the issue and providing a precise problem formulation together with appropriate empirical results is a worthy contribution in and of its own.

**Claims And Evidence:**

The paper provides convincing support for the theses put forward by demonstrating its results both theoretically and empirically.
The paper provides strong evidence for the observation that verifiers must do a better job at tieing their guarantees to the assumed implementation.

**Essential References Not Discussed:**

No.

**Experimental Designs Or Analyses:**

The experiments underscore the theoretical contribution of the paper.
I appreciate the evaluations of accuracy w.r.t. 8 different execution environments which nicely underscore the paper's point.

**Methods And Evaluation Criteria:**

The chosen methodology and evaluations are very convincing.

**Other Comments Or Suggestions:**

Proposition 5.1.:
Missing word: "Let [...] **be** the interval evaluation"

Page 5:
Typo: "is a suitable choice **for** bounding"

**Other Strengths And Weaknesses:**

The paper delivers some hard, but necessary lessons for the NN verification community at large:
It is paramount that NN verifiers put more effort into grounding their verification results w.r.t. implementation assumptions.
Proposition 5.1 and 5.4 might provide some answers to this end:
When fixing the execution order at execution, there might still be hope.
To this end, it is worth noting that NN verification is of particular interest in control applications where NNs run in embedded systems.
In such cases it should be possible to predetermine the execution order.

This does not lessen the important contribution of this paper, which underscores the importance of this grounding with a striking clarity.

**Questions For Authors:**

See "Theoretical Claims"

**Relation To Broader Scientific Literature:**

The paper correctly points out that the cited NN verification tools lack a principled treatment of (deployment-level) floating point behaviour.
It may be of interest to the authors, that this issue is also discussed in [ESOP25] as the "Implementation Gap" in NN verification.
Beyond the NN verifiers discussed in the paper, there also exists a line of work on quantized NN verification (e.g. [AAAI24] and implementation synthesis (e.g. [EMSOFT23]) which may provide partial answers to the problems discussed in this paper.

Overall the paper does a good job at reviewing relevant literature.

[ESOP25] https://arxiv.org/abs/2501.05867
[AAAI24] https://ojs.aaai.org/index.php/AAAI/article/view/30108
[EMSOFT23] https://dl.acm.org/doi/full/10.1145/3609118

**Theoretical Claims:**

The arguments for the theoretical propositions are sound in my understanding.
Only concerning the proof of Proposition 5.5 I have a question:
For the multiset {$1.1,1.1,1.1,1.1,1.1,1.1,1.1,1.1,1.1,\omega$} you state that $U_r=\omega+18$.
If I am understanding this correctly, because the smallest number after $\omega$ is $\omega+2$ we make a step of *at least* $2$ for each addition (because the floats become less and less dence), but could it not be the case that $U_r$ is larger than $\omega+18$?
This does not invalidate the proof of course.

---

> ### Author Rebuttal · Authors · 2025-03-29
>
> We thank the reviewer for the thorough and positive review and the many useful suggestions and pointers! We will incorporate these in the paper. While we agree with all of the comments, we elaborate on some of the specific points.
>
> **Proof of Proposition 5.5.** Indeed, if the number of bits of the floating point representation is very small, $U_r$ may become even larger. For example, when using the 8-bit minifloat format, $\omega=16$, so adding the 9th 1.1 will increase the sum by 4, resulting in $U_r=\omega+20$ (which, indeed, would not affect the validity of the proof). However, we mention in the appendix that we consider IEEE 754 representations. We made this more precise by stating that the proofs assume at least IEEE half-precision (or binary16) format, which is the lowest resolution IEEE 754 standard float. For a smaller number of bits and other alternative floating point definitions, the proofs are still valid with slight adjustments.
>
> **Recommended references.** We found the three included references very interesting and relevant, and we plan to incorporate them into our work. The ESOP 2025 accepted paper (parallel to our work) is particularly interesting as it also studies issues that stem from the difference between idealized descriptions and actual implementations. The paper is complementary to our work, approaching the problem from a different angle by focusing on the need for more accurate specifications for verification. We also agree that quantized neural networks (the remaining two references) are a promising approach if one wants to get rid of the floating point issues in verification, although it requires non-trivial future work to assess the trade-offs and limitations, as well as the potential remaining floating point issues involved in the quantization process itself.

---

### Official Review · Reviewer_pP6j · 2025-03-13

**Overall Recommendation:** 4

**Summary:**

The paper discusses a crucial problem with neural network verification: The networks are evaluated not as pure mathematical functions, but on real-world hardware that depends on specific floating point precision and computation orderings. However, this fact is often not taken into account by NN verification tools. Therefore, they may state that a network is robust to a certain perturbation, even though adversarial examples exist due to rounding errors.
The authors state this property both in general, and demonstrate it in practice. They show that all tested verification tools give unsound results for specifically crafted adversarial networks.

# Update after Rebuttal
Thank you for your rebuttal. Based on your responses, as well as the clarifications to the questions raised by the other reviewers, I will increase my rating to "Accept". I recommend to spend part of the extra page that the camera ready version is allowed to have on further explaining why your contribution is important: Soundness cannot be restored by using arbitrary-precision verifiers, and the usually considered verification problem differs from the deployed version. This seems to be the biggest criticism raised.

**Claims And Evidence:**

The claim of the paper is demonstrated in practice on a set of experiments. The authors successfully generate network architectures that are incorrectly evaluated by a number of verification tools. Therefore, they are able to show that floating point errors are currently not correctly handled in existing work.

**Essential References Not Discussed:**

I am not aware of specific literature that is missing.
However, floating point arithmetic must be of interest to e.g. verification of C programs, too. How do those verifiers handle these issues? E.g., what would happen if the adversarial network is converted to a simple C program (by enumerating all operations for each neuron) and verified? The paper may benefit from looking into this in more detail.

**Experimental Designs Or Analyses:**

The experiments appear to be sound.

**Methods And Evaluation Criteria:**

The evaluation makes sense: They demonstrate the theoretical risk for unsoundness exists in practice by evaluating multiple verification tools on a number of adversarial networks. This succeeds at proving their point that existing verification tools are unsound for specific floating point rounding errors.

**Other Comments Or Suggestions:**

N/A

**Other Strengths And Weaknesses:**

While the main point (tools not being sound w.r.t. floats) is not novel, the technique to design the adversarial network is. Its strength is demonstrated by the experimental evaluation, where it can fool all evaluated verifier. Therefore, I consider the paper to be original and of significance.

**Questions For Authors:**

1) Is proposition 5.1 supposed to use $f([x, x])$? This would apply $f$ to the interval only containing $x$, which I did not expect here.
2) See my question in "Essential References Not Discussed": How do verifiers for other languages solve the floating point problem, e.g. for C programs? Could you fool them, too, using your adversarial architecture? If not, how do they solve this, and could NN verification tools implement similar safeguards?

**Relation To Broader Scientific Literature:**

The fact that verification tools may not be sound in the face of floating point errors was known, but not evaluated to the extend done in this paper. Some tools choose to ignore this issue completely, others claim to be sound to floating point errors. This paper does a good job at demonstrating the risks, and describing how to test future verification tools for soundness w.r.t floating point arithmetic.

**Theoretical Claims:**

I did not check the correctness of the proofs.

---

> ### Author Rebuttal · Authors · 2025-03-29
>
> We are happy to see that the reviewer considers the paper original and of significance. Let us address the issues raised.
>
> **Issue 1: The main point is not novel.** The reviewer correctly states that floating point issues have been shown to fool certain approaches to verification. However, some verifiers were claimed to be sound, and indeed, they cannot be fooled with the known approaches because they carefully consider floating point representation and computation, although *only w.r.t. the idealized verification problem we present in Eq (1) but not in the deployed verification problem in Eq (2)*. So, we consider the theory behind our adversarial networks a novel contribution, shedding light on the specific issues with deployed neural networks.
>
> **Question 1**: Yes, the formulation is correct. The theoretical value $f(x)$ (thus, evaluated on a single input $x$) might not be representable, even if $x$ is representable in the given floating point representation. Therefore, even on a single representable input, IBP will compute an interval $[a,b]$ where both $a$ and $b$ are representable and, in general, $a<b$.
>
> **Question 2**: Numeric program verification is indeed a very similar problem (note also the ESOP 2025 reference included by reviewer *Twzp* that addresses exactly this idea). Accordingly, sound verifiers are frequently based on classical techniques and ideas (Miné, 2004). We are not aware of any works in the literature that would offer solutions to the problems we raised in the general case. If the program is deterministic (no parallelization, fixed floating point representation, and execution order), then the expression tree is fixed, and approaches like IBP will be sound. However, if there is true randomness (that is, not pseudo randomness), like a random ordering of some associative computations based on the current state of the environment, then verifying the program will be much harder. In the case of neural networks, where large associative expressions dominate in an inherently random environment due to paralllelization, hardware, load balancing, etc., this issue is central for provably sound verification.

---

> > ### Comment · Reviewer_pP6j · 2025-04-03
> >
> > My previous comment regarding my score increase and recommendation for the final paper was supposed to be the rebuttal comment that I'm asked to submit.

---

### Official Review · Reviewer_BYBy · 2025-03-14

**Overall Recommendation:** 2

**Summary:**

This paper studies the problem of the gap between theoretical soundness and practical soundness of neural network verification, which is commonly seen in the deployment of neural networks. It also proposes adversarial networks based on such characteristics to fool the verifiers to compromise soundness. Experiments validate the findings of such unsoundness.

## update after rebuttal

Thanks for the further clarification of the setting and the problem the paper is tackling. I feel like it is a borderline paper, and I am a bit conservative because the claims and findings in the paper are serious and are supposed to need more empirical results to justify them. I would like to give a score of 2.5 if it is allowed.

**Claims And Evidence:**

It is quite an interesting topic for neural network verification. However, it seems that the key findings are due to floating-point issues, which is not very impressive and surprising.

**Essential References Not Discussed:**

References look good to me

**Experimental Designs Or Analyses:**

- One major concern regarding the experiments of adversarial networks lies in that it seems to be very easy to "defend" such attack, because it typically has larger weights or abnormal memory costs during inference.

- Besides, since floating error can influence the general performance of neural networks [1], I wonder how the adverial networks perform regarding the clean accuracy, i.e. the normal and general performance. If the clean accuracy is also compromised due to such patterns, it is actually not harmful to verifiers because such neural networks themselves are flawed and there is no need to verify them.

[1] Li et al. Reliability Assurance for Deep Neural Network Architectures Against Numerical Defects, 2023

**Methods And Evaluation Criteria:**

- In Section 4.2, verified domain and verified property adopt different assumptions regarding the rounding issues, which seems to be not fair and needs more justification.

- I agree the IBP suffers floating error because the bounds are concretized at each layer. However, in symbolic bound propagation verification methods (e.g. CROWN), the floating error seems to be intuitively alleviated because  the bounds  are concretized only at the last layer.

**Other Comments Or Suggestions:**

See above

**Other Strengths And Weaknesses:**

See above

**Questions For Authors:**

See above

**Relation To Broader Scientific Literature:**

Related to neural network verification

**Theoretical Claims:**

Yes, looks good to me.

---

> ### Author Rebuttal · Authors · 2025-03-29
>
> We are pleased to read that the reviewer finds the topic interesting. Let us discuss the main issues raised here.
>
> **Issue 1: The discovered issues are not surprising.** Please refer to our answer to reviewer *N1Xq* that covers the same issue in detail.
>
> **Issue 2: In symbolic methods, the floating point issue is alleviated.** We specifically discuss symbolic approaches in Section 5.4, where we prove that they do not guarantee sound verification. This has to do with the fact that the deployment-related issues we study represent a problem even when computing the sum of inputs (arguably, the simplest possible function), as we explain at the beginning of section 5.1. In the case of the sum, symbolic approaches become essentially equivalent to IBP, regarding floating point error. We discuss them separately only because some specific symbolic methods use a slightly safer method to bound intervals, but we prove that this does not suffice. Therefore, unfortunately, symbolic approaches do not alleviate this specific problem.
>
> **Issue 3: The Definitions in Section 4.2 need more justification.** Since the discussion of the verified property does not make any assumptions, we believe the reviewer means the definition of the verified domain, where we use the definition $D_{p,\epsilon,{\cal E}}= D_{p,\epsilon}\cap X$, while in reality this is not perfectly correct, because this definition does not take into account that the p-norm cannot be computed exactly either, which in turn might make some different deployed domains (based on the exact p-norm) indistinguishable for certain verifiers. However, this slight simplification does not affect the validity of our theoretical results, because we study only single-point inputs (see also our answer to *Question 1* of reviewer *pP6j*).
>
> **Issue 4: Abnormal weights and memory costs make the attacks easy to detect.** Our backdoored networks do not need abnormal amounts of memory. As for weights, we stress that our focus was not to provide undetectable attacks, instead, we wanted to provide existential proofs for the theoretical observations we made (namely, that none of the known verifiers are practically sound). However, the backdoors we used can be made more undetectable via distributing the operations over many neurons and many layers so that no individual weights are too large, as was done, for example, in (Zombori et al 2021). Such optimizations are out of the scope of the paper.
>
> **Issue 5: Clean accuracy might be affected by the attack.** Please recall that our backdoor design is not based on training on poisoned data; instead, we modify a trained network. Our backdoors are activated under certain circumstances (precision or operation order), depending on the deployment environment. Our simplest backdoor, for example, functions as a switch that is activated by floating point precision: the backdoor is active in 32-bit environments and inactive in 64-bit environments. When the backdoor is inactive, it does not affect the original clean network, thus, accuracy will stay the same. When the backdoor is active, it changes every prediction of the network by shifting the class label, which drastically reduces accuracy.
>
> We thank the reviewer for reference [1], which is an interesting paper, but it focuses on a rather different problem (detecting potential numeric errors (Inf, NaN) in the entire input domain as opposed to bounding error in a limited (safe) domain), besides, it uses IBP as well to examine numeric error, and so it does not take into account the deployment environment either.

---

> > ### Comment · Reviewer_BYBy · 2025-04-03
> >
> > Thank you for the detailed clarification. It is indeed an interesting topic, however, I am still concerned about the lack of empirical justification to support the abnormal memory and clean accuracy with the modification of the trained network. Since this paper is to tackle the issues under the deployment environment, these empirical justifications are of more importance, and the non-sound claim needs more experimental evidence beyond discussion. I will keep my score as it is borderline, and more work can be done to strengthen the paper.

---

> > > ### Author Response · Authors · 2025-04-04
> > >
> > > Thank you for the comment. We feel that our answers to issues 4 and 5 were not formulated well enough, so we use this last opportunity to try and provide an even more informative answer.
> > >
> > > On the point of **clean accuracy**, let us try a different angle, reacting directly to the comment
> > >
> > > > If the clean accuracy is also compromised due to such patterns, it is actually not harmful to verifiers because such neural networks themselves are flawed and there is no need to verify them.
> > >
> > > One way to interpret this is assuming that the reviewer is suggesting that if a network is so bad that it is obviously useless then we need not verify it in the first place. True, but our networks are not obviously useless. The backdoor is activated only in certain environments, and the backdoored network passes verification, so it looks like a safe network. Also, in most environments, the backdoored network works well (when the backdoor is not active). What the verifier misses is that there are some environments in which it does not work well. This is, in fact, our main message. Verification should give us a guarantee **in advance** that the network will be fine **everywhere** it is going to be applied.
> > >
> > > It seems we did not understand the reference to abnormal memory usage, and unfortunately, we still don't. Indeed, the backdoored networks are slightly larger than the original networks due to adding the backdoor, but the number of extra parameters is relatively small and practically independent of the size of the original network. Also, this is irrelevant because our goals do not include creating networks indistinguishable from some reference network. Large weights are involved, too; however, as stated in our original rebuttal, this is not a problem here because our example networks still serve the purpose they were designed to serve and, as we mentioned, one could get rid of large weights with known techniques, but it was not the goal here either.
> > >
> > > We hope we were able to clarify at least some of the issues.

---

### Official Review · Reviewer_N1Xq · 2025-03-22

**Overall Recommendation:** 2

**Summary:**

The paper discusses the soundness of neural network verifiers. In particular, it exploits the order of floating point operations which may lead to unsound bounds computed by a neural network verifier.

**Claims And Evidence:**

The paper wants to demonstrate the most current neural network verifiers cannot fully handle the floating point errors introduced in a neural network. The paper constructs a few networks which contain a backdoor triggered by floating point erros and then evaluates whether existing verifiers produce unsound results on these inputs.

**Essential References Not Discussed:**

During the discussion period, I realized that the authors did not sufficiently discuss existing efforts from the formal methods community on dealing with floating point soundness. To improve the contributions of this paper, the best way is to discuss how to address these challenges in a practical way, specialized to the neural network verification settings. I realize that the paper did not discuss related works dealing with floating point soundness issues in formal verification, which may have caused the misinterpretation. I recommend these papers, which may help the authors to see what have been done on addressing this well-known issue and how they can be specialized to the NN verification setting.

A Two-Phase Approach for Conditional Floating-Point Verification: https://mariachris.github.io/Pubs/TACAS-2021.pdf
Scalable yet Rigorous Floating-Point Error Analysis: https://shemesh.larc.nasa.gov/fm/papers/SAFECOMP2017-draft.pdf
Correct Approximation of IEEE 754 Floating-Point Arithmetic for Program Verification:https://shemesh.larc.nasa.gov/fm/papers/SAFECOMP2017-draft.pdf
Floating-Point Verification using Theorem Proving: https://www.cl.cam.ac.uk/~jrh13/papers/sfm.pdf
Automatic Estimation of Verified Floating-Point Round-Off Errors via Static Analysis: https://shemesh.larc.nasa.gov/fm/papers/SAFECOMP2017-draft.pdf
Formal Verification of Floating-Point Programs: https://www.lirmm.fr/arith18/papers/filliatre-formal.pdf
FORMAL VERIFICATION OF AN IEEE FLOATING POINT ADDER: https://www.df7cb.de/cs/publications/2001/fpadder-cb.pdf
Automating the Verification of Floating-Point Programs: https://inria.hal.science/hal-01534533/document

**Experimental Designs Or Analyses:**

Several neural network verifiers are tested against the different types of backdoored networks. Not surprisingly, many of the verifiers do not consider the floating point ordering issues in their implementation, and may output unsound bounds given the network prepared by the author.

**Methods And Evaluation Criteria:**

The basic technique used in this paper is to construct certain neurons as a “backdoor” whose calculation involves large numerical errors and then be detected to trigger a different behavior of the network. To evaluate the soundness of existing verifiers, some neural networks are constructed in this way and then verified using these verifiers to see if they are sound against such backdoors. If they claim the network is “safe”, then they are unsound.

**Other Comments Or Suggestions:**

The usability and soundness tradeoff is well-known when applying formal methods - for example, when you verify a C program, do you want to verify down to the bit or gate level? Or just at the semantic level? The hardware (e.g., floating point unit) may have rounding errors during execution, and do we want to verify down to the logic gate level to ensure the correctness of the program? Technically, it is possible since all the actual circuits to implement these computations (whether it is a general computer program or a neural network) are implemented using logic gates, so the entire problem is a boolean satisfiability problem. If you verify exactly the hardware you execute your neural network, then there is no soundness issue. However, it is often not practical because of scalability of verifying to this level (floating point addition, multiplication etc result in very complex logic) and the diversity of hardware implementation. Instead, most neural network verifiers work at the semantic level, assuming the underlying hardware is accurate. By crafting artificial examples that exploit these assumptions, surely you will see the soundness issues reported in this paper. So I personally feel this result is not novel enough for publishing alone - more contributions, such as how to address this challenge, or demonstrate unsoundness in neural networks in production, will be helpful.

**Other Strengths And Weaknesses:**

The finding of this paper is interesting, although it is not very surprising - any numerical solvers will have numerical errors, and give ill-conditioned inputs, we expect them to fail. I am also not sure about its practical importance - the constructed cases in this paper are quite artificial and can hardly be linked to real networks being verified, and the methodology is also straightforward. If such a soundness issue can be discovered in a more realistic scenario, the paper could have a better impact.

**Questions For Authors:**

Technically, we could implement a verifier to consider the worst-case floating point error but that will significantly reduce the scalability of neural network verifiers. This is currently not done since the biggest challenge in neural network verification is still the scalability to handle large networks, rather than floating point soundness. It’s always a tradeoff - just like in program verification, we don’t always verify down to bit-level accurate floating point numbers - it is simply too slow; yet, verifiers are still useful despite the fact they theoretically could be unsound. It is worthwhile to discuss how to efficiently improve floating point soundness in a verifier while maintaining its efficiency.

**Relation To Broader Scientific Literature:**

Neural network verification is an important topic for ensuring provable guarantees of neural networks in mission-critical systems. The soundness of neural network verifiers is the cornerstone of provable guarantees of neural networks, and is an important topic to study.

**Theoretical Claims:**

The paper made some theoretical discussions on the floating point errors under standard IEEE representations. These claims look reasonable.

---

> ### Author Rebuttal · Authors · 2025-03-29
>
> We are encouraged by the fact that the reviewer considers the paper interesting and the topic important.
>
> Before moving on to addressing the specific issues raised, let us recall that our specific interest lies in provably sound verification (a motivation shared by a sizable community) as opposed to heuristic verification.  This is important because this determines what is interesting and what is less so in this context.
>
> **Issue 1: The discovered issues are not surprising.** Our goal was to shed light on an issue that has been in a blind spot because, although most works on provably sound verification give full consideration to floating point representation, features of deployment environments such as the stochasticity of associative operations have not been in focus. We have to admit that initially, we did find it a bit surprising that none of the current verifiers claimed to be provably sound are actually sound in deployment. This motivated us to dive deeper and prove this theoretically, and to demonstrate this in practice as well.
>
> **Issue 2: Practical motivation is unclear.** As stated above, our primary motivation was the theoretical problem of provably sound verification in deployment. Nonetheless, we demonstrated the issues through backdoored networks that reveal that it is practically feasible for an attacker to circumvent any verifier just by having enough information about the deployment environment. Indeed, our examples are artificial, but the “naturalness” can be enhanced with more distributed designs that avoid extremely large weights and other obvious patterns. This was not the focus of this work.
>
> **Question: Is the efficient verification of provable soundness possible in deployment?** We do not think so, at least not in the general case. In fact, this is one of our intended takeaway messages. As we mention in the paper, for example, covering all the possible orderings is not likely to be computationally feasible at scale. In extremely sensitive environments where provable soundness is required, one needs to control the execution tightly, and with such extra assumptions, provable soundness in deployment might indeed be feasible, at least no less feasible than the verification of the theoretical model of the network.

---

> > ### Comment · Reviewer_N1Xq · 2025-04-02
> >
> > Thank you for the response and I appreciate the discussion.
> >
> > I still find the paper interesting but not very surprising since numerically ill-conditioned inputs are well-known problems for any solvers/verifiers. Practically, it is really a tradeoff between usability and soundness - although we could, for example, run any verification algorithm using soft float with arbitrary precision to avoid the issue discussed in this paper, it is just not worthwhile in practical scenarios.
> >
> > The paper can be improved by either demonstrating more realistic concerns in practical networks (rather than manually constructed ill conditions), or proposing efficient and realistic solutions to improve numerical stability specialized to the NN verification setting (addressing the usability and soundness tradeoff). I hope the authors can consider these directions, and I unfortunately cannot support acceptance for the current version of this paper.

---

> > > ### Author Response · Authors · 2025-04-03
> > >
> > > Thank you for your reply, which helped us identify an important misunderstanding. Please note that using arbitrary precision during verification does **not** help us avoid the problem we raised, only if the network is also evaluated in arbitrary precision during deployment. Our work raises the problem that verifiers do not take into account the deployment environment; that is, current verifiers work on ill-specified versions of the verification problem. This means that even if they are provably sound (and for that, one does not even need full precision), they are verifying the wrong thing. This gap between idealized and actual specifications (Eqs. (1) and (2)) can be practically exploited by attackers (as we demonstrate).
> > >
> > > Also, a soundness-usability tradeoff is tricky to define because soundness is a binary concept (either you have it or you don't). One could define a continuous version of soundness by, for example, saying that the larger the (possibly negative) proven lower bound of $f(x)$ is, the more sound the method. However, in our case, an attacker can change the behavior of the network **arbitrarily**, that is, if one ignores these problems in the name of usability, then there is **no soundness** at all in this continuous sense either.

---

### Decision · Program_Chairs · 2025-05-01

**Decision:**

Accept (spotlight poster)

**Comment:**

This paper attracted very diverse set of reviews from reviewers in different domains. The central claim of the paper is that "theoretical soundness (bounding the full-precision output while computing with floating point) does not imply practical soundness (bounding the floating point output in a potentially stochastic environment)."The authors provide a strong theoretical and empirical evidence for the claim.

The papers on soundness of verification and testing of system may always give effect of "lack of surprise in retrospect" -- and in some ways, that is bound to happen unless the paper uncovers a catastrophic issue (but then such papers are rarely expected in academia as that's more of issue of testing). The fact that the paper attracted passionate discussion among reviewers highlights the "interestingness" of the paper and therefore, certainly worthy of being a spotlight.